# Removal of Sulfamethoxazole, Sulfathiazole and Sulfamethazine in their Mixed Solution by UV/H_2_O_2_ Process

**DOI:** 10.3390/ijerph16101797

**Published:** 2019-05-21

**Authors:** Guangcan Zhu, Qi Sun, Chuya Wang, Zhonglian Yang, Qi Xue

**Affiliations:** School of Energy and Environment, Key Laboratory of Environmental Medicine Engineering of the Ministry of Education, Southeast University, Nanjing 210096, Jiangsu, China; sllvfx@163.com (Q.S.); wang-cy@seu.edu.cn (C.W.); yangzhonglian@seu.edu.cn (Z.Y.); xq0316xlr@163.com (Q.X.)

**Keywords:** sulfonamides, mixed solution, UV/H_2_O_2_ oxidation, photodegradation, advanced oxidation process

## Abstract

Sulfamethoxazole (SMZ), sulfathiazole (STZ) and sulfamethazine (SMT) are typical sulfonamides, which are widespread in aqueous environments and have aroused great concern in recent years. In this study, the photochemical oxidation of SMZ, STZ and SMT in their mixed solution using UV/H_2_O_2_ process was innovatively investigated. The result showed that the sulfonamides could be completely decomposed in the UV/H_2_O_2_ system, and each contaminant in the co-existence system fitted the pseudo-first-order kinetic model. The removal of sulfonamides was influenced by the initial concentration of the mixed solution, the intensity of UV light irradiation, the dosage of H_2_O_2_ and the initial pH of the solution. The increase of UV light intensity and H_2_O_2_ dosage substantially enhanced the decomposition efficiency, while a higher initial concentration of mixed solution heavily suppressed the decomposition rate. The decomposition of SMZ and SMT during the UV/H_2_O_2_ process was favorable under neutral and acidic conditions. Moreover, the generated intermediates of SMZ, STZ and SMT during the UV/H_2_O_2_ process were identified in depth, and a corresponding degradation pathway was proposed.

## 1. Introduction

The ubiquitous occurrence of pharmaceutical compounds as well as their conversion products and metabolites have been recognized as a rising environmental issue of global concern because they are widely and increasingly used for human and veterinary therapeutic purposes, and are constantly released into the aqueous environment. Antibiotics are a large family of emerging pseudo-persistent contaminants that pose adverse effects on the society and environment due to their antibacterial properties and biological activity [1]. Even at very low concentration, the antibiotics can still lead to an increase of bacterial resistance against antibiotics, and these resistant genes are usually persistent [1,2].

Sulfonamide is a collection of important synthetic sulfanilamide derivatives and is widely used in human and veterinary medicine [3,4,5]. It is reported that there are eight common sulfonamides including sulfamethoxazole, sulfamethizole, sulfadiazine, sulfacetamide, sulfisoxazole, sulfanilamide, sulfasalazine and sulfadoxine [6]. In this work, sulfathiazole (STZ), sulfamethoxazole (SMZ) and sulfamethazine (SMT) were chosen as the targets because: (a) given that these molecules are polar, amphoteric, water-soluble substances with light and thermal stability [1,7], they possess high migration ability and can easily and quickly spread in the environment [1,7]; (b) SMZ, SMT and STZ are the most heavily reported drugs in the surface water of China and exhibit a high pollution level [8,9,10]; and (c) all the sulfonamides are found in the wastewater treatment plant effluent, groundwater, surface water and even drinking water supply [5,10]. The detection of sulfonamides in the treated drinking water and wastewater treatment plant effluent indicates that conventional water and wastewater treatment processes cannot effectively eliminate sulfonamides. For these aforementioned reasons, it is highly desired to develop reliable water treatment methods that can efficiently remove sulfonamides at a trace level, especially in China.

In recent years, advanced oxidation processes (AOPs) have attracted more attention as complementary approaches to traditional water treatment or as alternative treatment methods before the discharging of industrial wastewater. Besides, with the generation of highly reactive and nonselective electrophiles such as hydroxyl radicals, AOPs can remove numerous nonbiodegradable organic pollutants and residual organic pollutants at a trace level in a wide range of water compositions [11]. The photolysis, photocatalysis, H_2_O_2_-enhanced photolysis, ozonation, electrochemical oxidation and Fenton are frequently studied AOPs [1,6,11,12,13]. Among these AOPs, the UV/H_2_O_2_ process consists of a combination of two well-known processes, direct UV photolysis and peroxidation, that lead to the generation of hydroxyl radicals, which make this process simpler and more effective for the removal of organic contaminants compared to peroxidation or photolysis respectively [14,15]. It is reported that some sulfonamides, such as sulfaquinoxaline sodium [14], sulfamethoxazole [16], sulfamethazine and sulfapyridine [17], can be efficiently removed by the UV/H_2_O_2_ process, and the photochemical oxidation of a single antibiotic with a high concentration (>10 mg L^−1^) was elucidated. However, other contaminations in the water may dramatically affect the removal of the target pollutant by competitively interacting with hydroxyl radicals and photons or with each other among these compounds. The study about the oxidation of antibiotics with a lower concentration in the mixed solution using the UV/H_2_O_2_ process is quite limited.

In this work, the photochemical oxidation of SMZ, STZ and SMT in their mixed solution using the UV/H_2_O_2_ process was innovatively explored. The effects of various parameters including the initial concentration of the mixed solution, UV light intensity, H_2_O_2_ dosages and the initial mixed solution pH on the removal of target antibiotics were evaluated. Based on the identification of the generated intermediates and the proposing of the corresponding degradation pathway, the feasibility of using the UV/H_2_O_2_ process for the degradation of sulfonamides was demonstrated.

## 2. Materials and Methods

### 2.1. Chemicals and Reagents

SMZ, STZ and SMT were purchased from Sigma-Aldrich, USA. The purity of all the above standards was >98%. The HPLC grade acetonitrile was purchased from Supelco Co. USA. The formic acid used in the high-performance liquid chromatography was guaranteed reagent. The hydrogen peroxide (H_2_O_2_, 30% w/w) was obtained from Sinopharm Chemical Reagent Co. Ltd. (Shanghai, China). The analytically pure hydrochloric acid (HCl) and sodium hydroxide (NaOH) were used to adjust the pH. The ultrapure water produced by the Milli-Q water purification system was used to prepare all solutions.

### 2.2. Experimental Procedures

The photochemical experiments were carried out in a 2.5 L cylindrical glass sleeve reactor (360 mm height, 100 mm diameter) at room temperature (25 ± 2 °C). The reactor was covered with aluminum foil to eliminate the effects of natural light and avoid the exposure of UV radiation. The reactor was operated in batch mode and a magnetic stirrer device was placed at the bottom of the reactor to provide a rapid mixing. A low-pressure mercury UV lamp emitting at 254 nm was inserted in the center of the cylindrical reactor and protected in a quartz sleeve (outer diameter 25 mm, length 305 mm). In all cases, the water sample containing a mixture of the three compounds was added to the reactor. The initial concentration of the mixed solution of 100 μg L^−1^ was set. The predetermined concentration of hydrogen peroxide was added to the reactor and immediately subjected to UV irradiation to start the reactions. After initiating the reactions, samples were collected at regular intervals, and the concentrations of antibiotics were measured using a high-performance liquid chromatography (HPLC). The involved H_2_O_2_ solution and sulfonamide solution were prepared at the time of use. For comparison, the direct photolysis experiment (no H_2_O_2_ was added into the reactor) and control experiment with H_2_O_2_ (without UV irradiation) were conducted in the same conditions. A series mixture solution with different concentrations of sulfonamides ranging from 50 μg L^−1^ to 200 μg L^−1^ was used to investigate the influence of initial mixed solution concentration. In order to determine the effect of UV light intensity, the specific intensity was set as 5, 10 and 15 W respectively. Series concentrations of H_2_O_2_ (0, 30, 55 and 100 mg L^−1^) was selected to evaluate the influence of H_2_O_2_ concentration. Three initial solution pH values (5.0, 7.0 and 9.0) were used to determine the influence of initial solution pH. The pH of other water samples was not adjusted except for the experiments focusing on the effect of pH on the sulfonamides degradation. All experiments were performed three times with independent replication of data (*n* = 3).

### 2.3. Analytical Methods

The antibiotics concentration was analyzed using a HPLC (Agilent 1100) equipped with a UV detector and a C18 reverse-phase column (2.1 × 50 mm, 1.7 μm). The mobile phase was an 80:20 (v/v) mixture of acidified water (0.1% formic acid) and acetonitrile at a flow rate of 0.2 mL min^−1^. The UV detector wavelength was set as 265 nm. The injection volume and column temperature were 10 μL and 25 °C, respectively. The collected water samples during the experiment were directly used for the determination of residual target compounds by HPLC according to the above conditions. The pre-prepared target compound solution of different concentrations was used to create the calibration curve, which was employed to calculate the concentration of residual target compounds. Under this test condition, the detection limit of the selected sulfonamides was 5 μg L^−1^, and the relative standard deviations (RSD) of the three substances were all less than 0.2% to meet the detection requirements. The pH of the solution was measured using a PHSJ-4A pH meter. The UV-Vis spectra of the samples were obtained on a UV-1280 spectrophotometer (Shimadzu, Japan). The identification of the generated intermediates during the degradation of sulfonamides in the UV/H_2_O_2_ process was performed using an Agilent 1100 LC equipped with an Agilent 6530 Series Triple Quadrupole mass spectrometer (LC/Q-TOF-MS) system. The compound was ionized in the electrospray ionization (ESI) and operated in positive mode. The selected ion monitoring (SIM) mode and a scan range from 40 m/z to 3200 m/z was used to obtain the spectra of sulfonamides and their intermediates. The injection volume (20 μL) was conducted using an autosampler at a flow rate of 1 mL min^−1^ through an XDB-C18 (4.6 cm × 50 mm × 1.8 μm, Agilent, USA) column. The mobile phase was a mixture of acidified water (0.1% formic acid) and acetonitrile and it was followed by a gradient elution mode: (1) maintain the 10% acetonitrile in 0–5 min; (2) acetonitrile proportion gradually increased to 100% within 50 min; (3) 100% acetonitrile holding for 3 min. The gas temperature and source voltage were 400 °C and 4.0 KV, respectively.

The degradation of sulfonamides was evaluated based on the following equation:(1)$$\mathrm{Sulfonamides}\;\mathrm{degradation}\;\mathrm{rate}\;\left(\%\right)=\frac{C_{0}-C_{t}}{C_{0}}\times 100\%$$
where *C*_0_ and *C_t_* indicate the concentration of sulfonamides at the beginning and at the given reaction time *t*, respectively.

In addition, in order to illustrate the degradation rate of UV/H_2_O_2_ process for the removal of sulfonamides quantitatively, the kinetics of sulfonamides degradation was investigated using the following pseudo-first-order reaction kinetic model:(2)$$\ln C_{t}=\ln C_{0}-kt$$
where *C*_0_ and *C_t_* represent the concentration of sulfonamides at the beginning and at the given reaction time *t*, respectively. *k* is the pseudo-first-order degradation rate constant which is obtained by the slope of the linear regression of the sulfonamides degradation data points.

## 3. Results and Discussion

### 3.1. Comparison of Absorption Spectra of Target Compounds

In order to investigate whether the UV light irradiation centered at 254 nm can degrade SMZ, STZ and SMT, the absorption spectra of SMZ, STZ and SMT were measured with the wavelength ranged from 200 to 400 nm, and the result is shown in Figure 1a. The SMZ has a strong absorption peak around 265 nm. However, STZ has two strong absorption peaks located at 258 nm and 284 nm, and those of SMT occur at 240 nm and 262 nm, respectively. Therefore, all these three antibiotics possess a strong absorption of the 254-nm UV light irradiation, indicating that all these three antibiotics could be degraded by the UV lamp at the main wavelength of 254 nm.

### 3.2. Decomposition of Sulfonamides in the Mixed System

Figure 1b–d shows the degradation of sulfonamides in the mixed system under different conditions. When only 55 mg L^−1^ of H_2_O_2_ was added into the reaction system, the degradation of sulfonamides was negligible. When only sole UV irradiation was performed, the degradation of sulfonamides was quite limited, which might be attributed to the high molar absorption coefficient of sulfonamides at 254 nm [14]. The degradation rate constant of SMZ, STZ and SMT was found to be 0.155 min^−1^, 0.044 min^−1^ and 0.016 min^−1^, respectively, when there was only sole UV irradiation in the system (Figure 1e). However, when H_2_O_2_ and UV irradiation were performed together, the degradation rate of sulfonamides was substantially enhanced compared to that of sole UV irradiation. At this time, the degradation rate constant of SMZ, STZ and SMT increased to 0.276 min^−1^, 0.178 min^−1^ and 0.130 min^−1^, respectively (Figure 1e). According to previous studies, the oxidation of organic compounds through UV/H_2_O_2_ process can be divided into three stages: direct photolysis, H_2_O_2_ oxidation and indirect photolysis [11,18]. As is shown in Figure 1b–d, the H_2_O_2_ oxidation did not affect the transformation of sulfonamides, while the decomposition of sulfonamides occurred under UV radiation or UV/H_2_O_2_ oxidation, suggesting that both direct photolysis and indirect photolysis contributed to the decomposition of sulfonamides. Similar phenomena were reported, for instance, the removal of sulfaquinoxaline sodium [14] and pharmaceutically active compounds (PhACs) [19] through the UV/H_2_O_2_ oxidation process. This result indicates that the UV/H_2_O_2_ process was favorable for the degradation of sulfonamides in the mixed solution.

### 3.3. Effect of Initial Concentration of the Mixed Solution on Degradation Efficiency

The initial concentration of the mixed reaction solution is an important parameter because the specific concentration in environment water system and treated wastewater are quite different. The experiments were carried out at UV light intensity of 5 W and the H_2_O_2_ dosage of 55 mg L^−1^. Figure 2 shows the effect of initial concentration of the mixed solution on degradation efficiency in the UV/H_2_O_2_ system. The degradation efficiency of SMZ decreased with the increase of the initial concentration of mixed solution in the range of test concentrations (Figure 2a). The SMZ degradation rate was 99.9% for the 50 μg L^−1^ mixed solution after 20 min of reaction, which decreased to 97.4% and 76.6% when the mixed solution concentration was increased to 100 μg L^−1^ and 200 μg L^−1^, respectively. The degradation pattern fitted the pseudo-first-order kinetic model, and the kinetic constant decreased from 0.352 min^−1^ to 0.197 min^−1^ and 0.076 min^−1^ respectively, when the initial concentration of the mixed solution increased from 50 μg L^−1^ to 100 μg L^−1^ and 200 μg L^−1^. Similar results were obtained when it comes to STZ degradation (Figure 2b) and SMT degradation (Figure 2c). In addition, it is worth noting that a nearly complete STZ, SMZ and SMT removal was achieved at the lowest concentration of 50 μg L^−1^. This result is meaningful because the specific concentrations in most ambient waters will not be more than the tested concentration in this range. The reduced reaction rate at a higher mixed solution concentration was attributed to competition between various sulfanilamide molecules and/or their intermediates formed during the oxidation reaction process [20]. At a higher concentration, the permeation of photons in the solution was reduced, which resulted in a lower concentration of hydroxyl radical. In addition, the sulfonamides and their converted products compete with the generated hydroxyl radicals in UV/H_2_O_2_ system [20,21,22]. Namely, the efficiency of sulfonamides removal was suppressed when the initial mixed solution concentration was enhanced.

### 3.4. Effect of UV Light Intensity on Degradation Efficiency

The effect of UV light intensity on the removal of sulfonamides is shown in Figure 3. The experiments were carried out at initial concentration of mixed solution of 100 μg L^−1^ and the H_2_O_2_ dosage of 55 mg L^−1^. In all investigated reaction times, the removal efficiency of sulfonamides substantially enhanced when the intensity of the UV light was enlarged. When the UV light intensity was set as 5 W, the degradation rate of SMZ was 17.1% and 99.2% for the reaction time of 1 min and 23 min, respectively. And the complete removal of SMZ occurred when the reaction time was more than 25 min. When the UV light intensity was enlarged to 10 W and 15 W, a considerable enhancement of the SMZ removal was achieved. A complete removal of SMZ was achieved at the reaction time of 12 min and 8 min for the UV light intensity of 10 W and 15 W, respectively. Meanwhile, the SMZ degradation rate constants greatly increased from 0.197 min^−1^ to 0.845 min^−1^ with the increase of UV light intensity from 5 W to 15 W. A similar trend was observed in the degradation of STZ and SMT. In the UV/H_2_O_2_ process, the hydroxyl radical was formed through the following equation [23]:(3)$$H_{2}O_{2}+hv\to 2\cdot \mathrm{OH}$$

The photolysis rate of H_2_O_2_ directly depended on the incident power. With a low UV light intensity, the photolysis of H_2_O_2_ was limited. With a higher UV light intensity, more hydroxyl radical was generated through the photodissociation of H_2_O_2_, and thus their removal rate was enhanced [22,23].

### 3.5. Effect of H_2_O_2_ Dosages on Degradation Efficiency

Figure 4a–c depicts the changes of sulfonamides degradation efficiency in a UV-irradiated reactor with different H_2_O_2_ dosages (0, 30, 55 and 100 mg L^−1^). In addition, the initial concentration of the mixed solution and the UV light intensity was set as 100 μg L^−1^ and 5 W, respectively. The degradation efficiency of sulfonamides enhanced with the increasing of the H_2_O_2_ concentration from 0 mg L^−1^ to 100 mg L^−1^. In the absence of H_2_O_2_, the photolysis of sulfonamides gave rather moderate results and led to a slow degradation of sulfonamides. When H_2_O_2_ was not added to the system, the removal efficiency of STZ, SMZ and SMT was 54.7%, 93.3% and 29.3% at the time of 20 min, respectively. The addition of 30 mg L^−1^ to 100 mg L^−1^ increased the STZ, SMZ and SMT degradation from 73.1%, 96.5% and 72.1% to 94.9%, 100.0% and 94.7% at a reaction time of 20 min, respectively. The degradation of sulfonamides at different H_2_O_2_ dosage fitted the pseudo-first-order kinetic models based on the Equation (2) and this model had a good fitness (R^2^ > 0.94) with the degradation of sulfonamides. When the H_2_O_2_ dosage was 0, 30, 55, and 100 mg L^−1^, the degradation rate constant of SMZ, STZ and SMT was 0.139, 0.171, 0.197, and 0.266 min^−1^, 0.039, 0.073, 0.137, and 0.181 min^−1^, and 0.014, 0.063, 0.101, and 0.158 min^−1^, respectively. It could be seen that the degradation rate constants of STZ, SMZ and SMT increased by 357.2%, 91.6% and 1043.5%, respectively, when the H_2_O_2_ concentration increased from 0 mg L^−1^ to 100 mg L^−1^. The H_2_O_2_ could absorb the photon energy from UV irradiation and the hemolytic splitting of the O–O bond in H_2_O_2_ resulted in the generation of activated hydroxyl radicals and atomic oxygen [12,20]. The enhancement of the H_2_O_2_ concentration would lead to the increase of energy harvesting and hydroxyl radical generation, and thus the oxidative destruction of sulfonamides was promoted.

In addition, the reaction rate constant as a function of H_2_O_2_ dosage is shown in Figure 4d. The increase of the H_2_O_2_ dosage resulted in the enhanced reaction rate constant for the degradation of STZ, SMZ and SMT, along with a robust linear relationship between H_2_O_2_ dosage and reaction rate constant (R^2^ > 0.96). This result indicates that the H_2_O_2_ essentially had the same effect on the reactions of these compounds in the range of tested concentration.

### 3.6. Effect of Initial Solution pH on Degradation Efficiency

The solution pH is recognized as one of the main factors that affect the photochemical reaction process [14]. The decomposition of sulfonamides during the UV/H_2_O_2_ process at pH of 5.0, 7.0, and 9.0 was conducted in this study. The initial concentration of the mixed solution, UV light intensity and H_2_O_2_ dosage was set as 100 μg L^−1^, 5 W and 55 mg L^−1^, respectively. It is clearly shown in Figure 5a–c that the degradation efficiency of SMZ and SMT gradually decreased when the initial solution pH increased. The degradation rate constant of SMZ and SMT decreased from 0.316 min^−1^ and 0.105 min^−1^ to 0.145 min^−1^ and 0.093 min^−1^, respectively, when the initial solution pH increased from 5.0 to 9.0. However, as opposed to the degradation of SMZ and SMT, the variation of initial solution pH improved the degradation rate of STZ. The degradation rate constant of STZ increased by 29.3% from 0.120 min^−1^ to 0.155 min^−1^ when the initial solution pH increased from 5.0 to 9.0. The effect of initial solution pH on the degradation rate of antibiotics during the UV/H_2_O_2_ process was caused by the competitive processes of hydroxyl radical generation and scavenging [24]. The hydroperoxide anion (HO_2_^−^) was present as a conjugate base of H_2_O_2_ under strong alkaline conditions [22]. When the pH increased, the proportion of HO_2_^−^ anion increased. The HO_2_^−^ was more reactive than H_2_O_2_ and the formation of hydroxyl radicals was enhanced under alkaline conditions [24]. Furthermore, H_2_O_2_ and hydroxyl radical could react with the HO_2_^−^ (Equations (4)–(6)) [14,25] and the reaction rate of the hydroxyl radical with the HO_2_^−^ was about 100 times higher than that with H_2_O_2_ [24,26].
(4)$$H_{2}O_{2}+{\mathrm{HO}}_{2}^{-}\to H_{2}O+O_{2}+{\mathrm{OH}}^{-}$$
(5)$$\mathrm{OH}+{\mathrm{HO}}_{2}^{-}\to H_{2}O+O_{2}^{-}$$
(6)$$\mathrm{OH}+{\mathrm{HO}}_{2}^{-}\to {\mathrm{HO}}_{2}^{\cdot }+{\mathrm{OH}}^{-}$$

This explains why the reaction rate was lower under strong alkaline conditions. In addition, the rate of self-decomposition of hydrogen peroxide increased with increasing the solution pH that also reduced the reaction rates, because the main decomposition product was oxygen and its oxidizing ability was lower than that of hydroxyl radical [14,24,26]. Besides, most sulfonamides contained two basic functional groups (group of aniline and heterocyclic base) and one acidic functional group (sulfonamide group), their ionization states were controlled by solution pH and acid dissociation constant pKa [21,27]. Thus, these two parameters might affect the degradation rate of sulfonamides. The pKa values of SMZ, STZ and SMT were shown in Table 1. They had a cationic form at pH < pKa_1_ and an anionic form at pH > pKa_2_ [16,21]. These different ionic states of sulfonamides might have a considerable impact on their reactivity and light absorbing properties. Previous study has shown that molar absorption coefficient of SMZ in different protonation states followed the order of protonation > neutral > deprotonation at wavelength below 300 nm [8]. From this perspective, it can also explain why SMZ was more resistant to UV degradation at a higher pH.

In order to intuitively investigate the effect of pH on the light absorption properties of sulfonamides, the UV spectra of selected sulfonamides at different pH conditions were determined. It can be clearly seen from Figure 5d–f that the change in pH had a considerable effect on the UV absorption spectra of the STZ, SMZ and SMT. The absorption peaks of the STZ, SMZ and SMT at pH = 2 were substantially weaker than those of pH = 6 and 11. When the pH increased, the UV absorption of SMZ and STZ exhibited a slight blue-shift, resulting in an increase of their molar adsorption coefficient at 254 nm. Macroscopically, the UV absorption of organic compounds at 254 nm was enhanced. The absorption peak of SMT gradually increased with the increase of pH. In general, in the wavelength range of 190–600 nm, the light absorption of organic matter was mostly related to the delocalized π electron system present in the structure. A change in pH would result in a reduction or even an elimination of the π electron delocalization effect, which would result in a change of the absorption characteristics and photolysis properties of an organic substance. Namely, sulfonamides possessed the delocalized π electrons and were subjected to the pH change. Therefore, in the absence of other organic substrates, the variation of pH caused changes in the light absorption characteristics of sulfonamides, and thus affected their UV degradation performance.

### 3.7. Identification of Sulfonamides Degradation Intermediates

The intermediate products of sulfonamides decomposition were analyzed using LC/Q-TOF-MS. The structure of transformation products and their MS fragmentation data in the UV/H_2_O_2_ process of SMZ, SMT and STZ were shown in Table 2, Table 3 and Table 4. SMZ, SMT and STZ had a retention time of 4.184 min, 3.984 min and 3.864 min, respectively, and they all yielded a [M+H]^+^ at m/z 254, 279 and 256. Based on the study of García-Galán et al. [17], the sulfonamides tend to lose the sulfonyl group under light irradiation. So the compound S1 at m/z 190.0182 was mainly a product of direct degradation by UV. The dihydroxy compound was identified at m/z 288.0661 (S3), which was caused by the hydroxyl radical attacking the double bond on the isoxazole ring. The formation of this compound could be explained based on the generation of a tertiary carbon-centered radical proposed by Hu et al. [28]. The compound S2 yielded an m/z ratio of 133.0609, which corresponded to the cleavage of S-N of S3. The compounds S4 yielded an m/z of 99.0562, for which the best-fit formula was C_4_H_6_N_2_O, which was recognized as a by-product of TiO_2_ photocatalysis [9]. The reason for this compound might be described as follows: a five-membered ring contained a double bond and a nitrogen atom, and the atoms in the five-membered rings were conjugated, which resulting in a weak bond attached to the nitrogen and sulfur bond. The compound S5 at m/z 216.0437 was formed by the opening of the isoxazole ring and the loss of one carbonyl group. The m/z of the S6 ion was higher than that of the parent compound, indicating that hydroxyl radical might create a SMZ molecule along with an intermediate. The S7 had the same m/z as the S6, but the retention times of these two compounds were quite different. One had a retention time of 4.049 min and the other was 2.882 min, indicating that these two compounds were not the same compound. Based on previous reports, it is possible to form two isobaric compounds by applying two fragmentation voltages to obtain “in-source” fragmentation [13]. The S7 presented an ion fragment at m/z 156.0116, which corresponded to the cleavage of the sulfonamide bond. This ion fragment together with the ion fragments of m/z 108.045 and 92.050 were also existent in the SMZ spectrum and most of the conversion products, indicating that the degradation of SMZ was owing to the attack of the hydroxyl radicals on isoxazole ring. In the S6 compound, the ion fragment at m/z 156 disappeared and an ion fragment at m/z 172.0065 appeared, suggesting that the hydroxyl radical attacked the benzene ring.

From the aforementioned discussion, the same degradation intermediates as SMZ were found during the degradation of SMT. The degradation intermediates of SMT in the UV/H_2_O_2_ oxidation process included: (1) S-N bond cleavage in SMT, producing S1 (m/z 140.0824) and S2 (m/z 124.0878), (2) the loss of sulfonyl group in SMT, generating the compound S3 (m/z 215.1301), (3) hydroxylation of the benzene ring in SMT, producing the compound S4 (m/z 295.0875), (4) the addition reaction of double bond at the six-member heterocyclic ring, generating the compound S1 (m/z 140.0824) and S7 (m/z 313.0537). In addition to the degradation products similar to SMZ, the degradation process also found the oxidation product of the amine group at the benzene ring (S5, m/z 293.0718). Similarly, the degradation products of STZ mainly included the S-N bond cleavage product (S1, m/z 101.0179), sulfonyl removed product (S2, m/z 226.0662), carbon-carbon double bond addition product (S4, m/z 290.0274), hydroxylation product (S5, m/z 272.0176) and the amino group oxidation product on the benzene ring (S7, m/z 270.0017).

Based on the identification results of intermediates in the degradation process of SMZ, SMT and STZ, the possible degradation pathways of sulfonamides through the UV/H_2_O_2_ process could be elucidated as shown in Figure 6. The contaminants present in the aqueous solutions might produce more toxic conversion products than the parent compound. Some researchers evaluated the potential environmental impact of sulfonamides and its byproducts [29,30,31]. They found that toxicity was higher at the beginning of the reaction and the toxicity reduced gradually with the further progress of the reaction. This showed that the intermediates of sulfonamides were degraded into the less toxic products as the reaction progresses. Therefore, the environmental impact of sulfonamides degradation intermediates was less than that before degradation.

## 4. Conclusions

To sum up, the decomposition of sulfonamides with a low concentration in their mixed solution through the UV/H_2_O_2_ process was investigated in depth. The degradation of each contaminant in the coexistence system fitted well with the first-order kinetic model. The degradation efficiency of SMZ, STZ and SMT in their mixed solution could be affected by multiple parameters including initial concentration of mixed solution, UV light intensity, H_2_O_2_ dosage and initial solution pH. An increase in UV light intensity and H_2_O_2_ dosage could improve the degradation of organic pollutants, while the increase of the initial concentration of the mixed solution would suppress the degradation efficiency. The degradation of SMZ and SMT via the UV/H_2_O_2_ process gradually decreased with initial pH, however, the degradation efficiency of STZ increased with the increase of the initial solution’s pH. The intermediates of SMZ, SMT and STZ in the UV/H_2_O_2_ process were further identified, and the possible transformation pathways of sulfonamides in the UV/H_2_O_2_ process were proposed in this work. The toxicity of sulfonamides was high at the beginning, and the toxicity was gradually reduced as the reaction progresses. This work exhibits the feasibility of the UV/H_2_O_2_ process for the degradation of organic pollutants in aqueous environment, which provides new insights for the selection of oxidants in water treatment processes.

## Figures and Tables

**Figure 1 ijerph-16-01797-f001:**
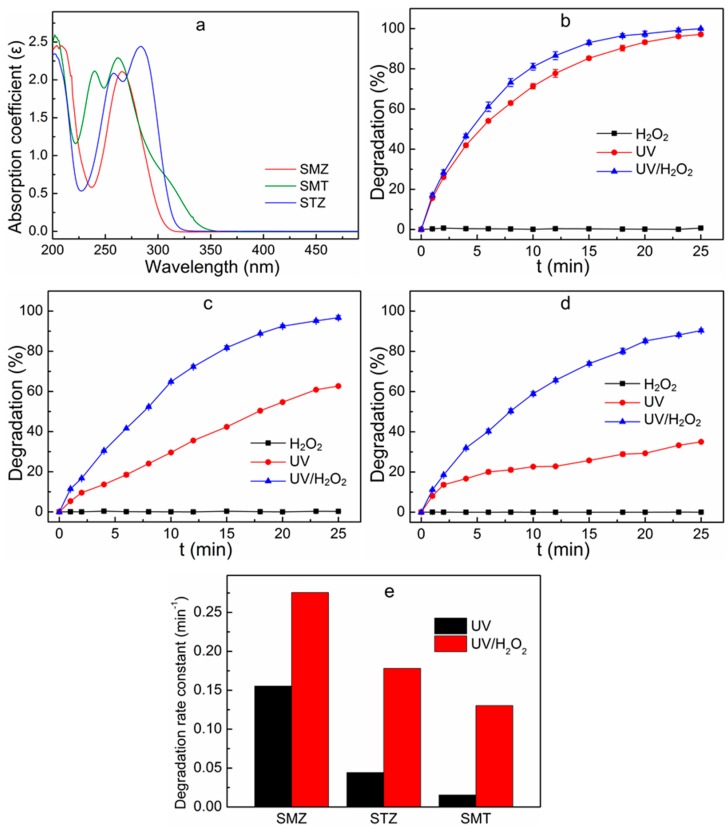
UV spectrum scans of sulfonamides (**a**), degradation of SMZ (**b**), STZ (**c**) and SMT (**d**) in different oxidation treatments, and degradation rate constants for sulfonamides degradation at different oxidation treatment (**e**). Reaction conditions: initial concentration of the mixed solution 100 μg L^−1^ (including SMZ 100 μg L^−1^, STZ 100 μg L^−1^ and SMT 100 μg L^−1^), UV light intensity 5 W, H_2_O_2_ dosage 55 mg L^−1^.

**Figure 2 ijerph-16-01797-f002:**
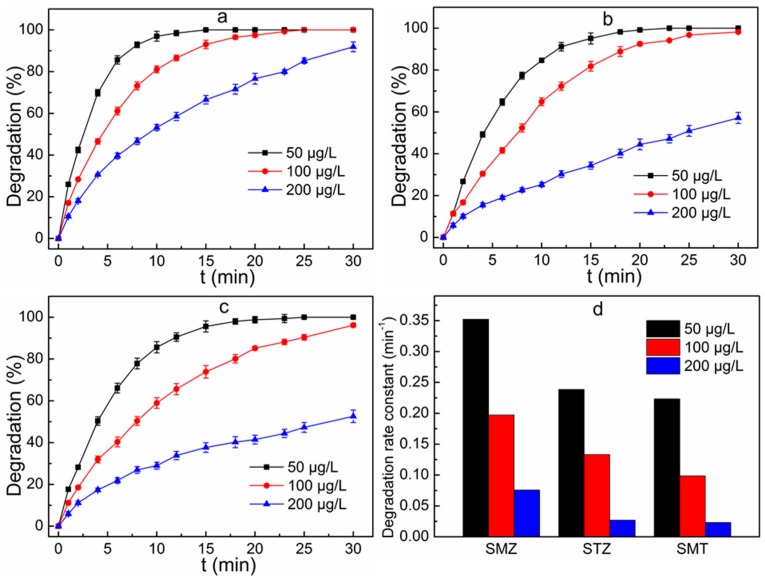
Effect of initial concentration of the mixed solution on the removal of SMZ (**a**), STZ (**b**) and SMT (**c**) and degradation rate constants for sulfonamides degradation at different initial concentration of the mixed solution (**d**). Reaction conditions: UV light intensity 5 W, H_2_O_2_ dosage 55 mg L^−1^.

**Figure 3 ijerph-16-01797-f003:**
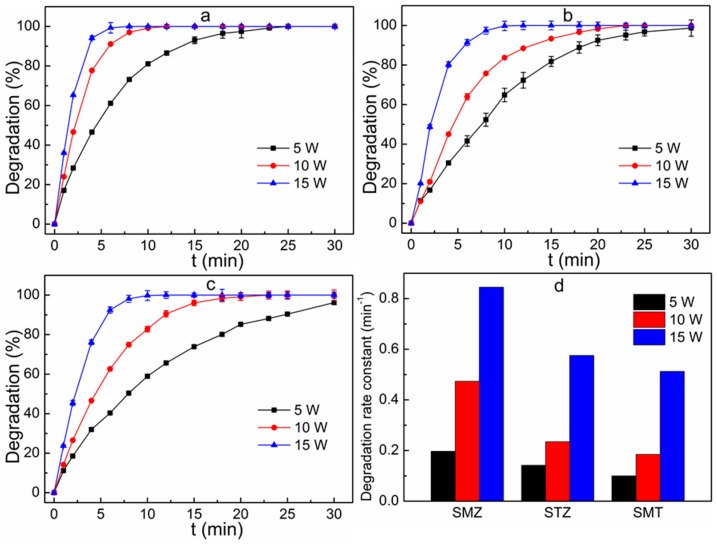
Effect of UV light intensity on the removal of SMZ (**a**), STZ (**b**) and SMT (**c**) and degradation rate constants for sulfonamides degradation at different UV light intensity (**d**). Reaction conditions: initial concentration of the mixed solution 100 μg L^−1^, H_2_O_2_ dosage 55 mg L^−1^.

**Figure 4 ijerph-16-01797-f004:**
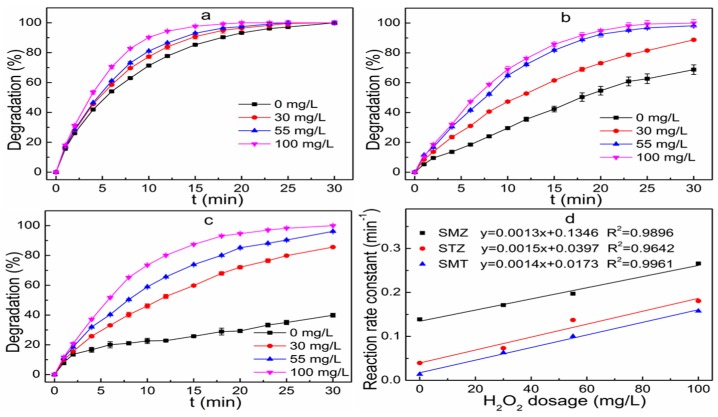
Effect of H_2_O_2_ dosage on the removal of SMZ (**a**), STZ (**b**) and SMT (**c**) and reaction rate constant as a function of H_2_O_2_ dosage (**d**). Reaction conditions: initial concentration of the mixed solution 100 μg L^−1^, UV light intensity 5 W.

**Figure 5 ijerph-16-01797-f005:**
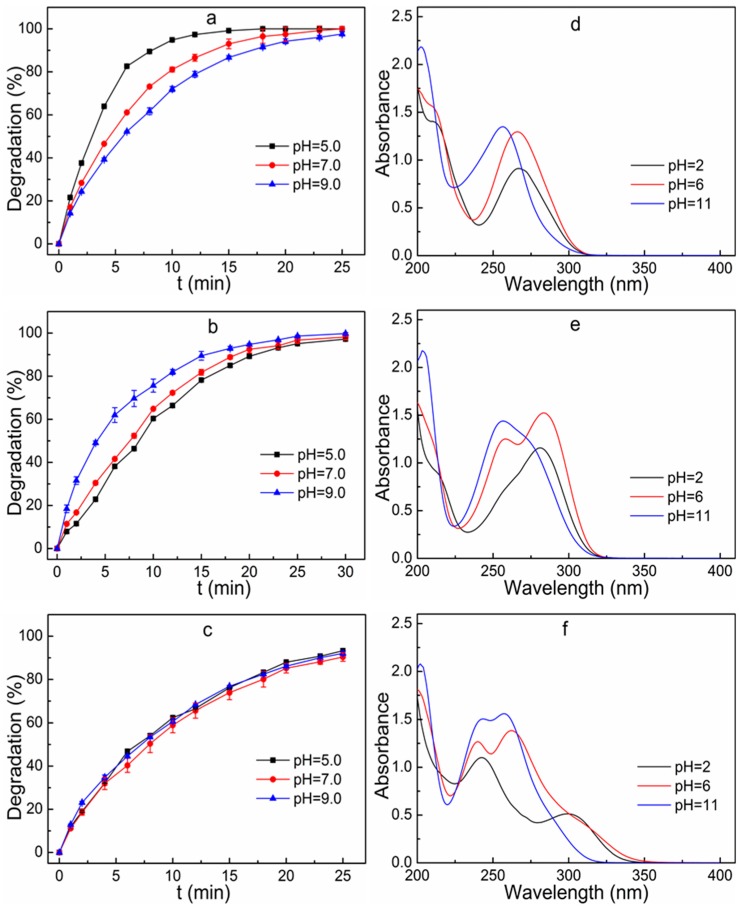
Degradation of SMZ (**a**), STZ (**b**) and SMT (**c**) in different pH and UV spectra of SMZ (**d**), STZ (**e**) and SMT (**f**) under different pH conditions. Reaction conditions: initial concentration of the mixed solution 100 μg L^−1^, UV light intensity 5 W, H_2_O_2_ dosage 55 mg L^−1^.

**Figure 6 ijerph-16-01797-f006:**
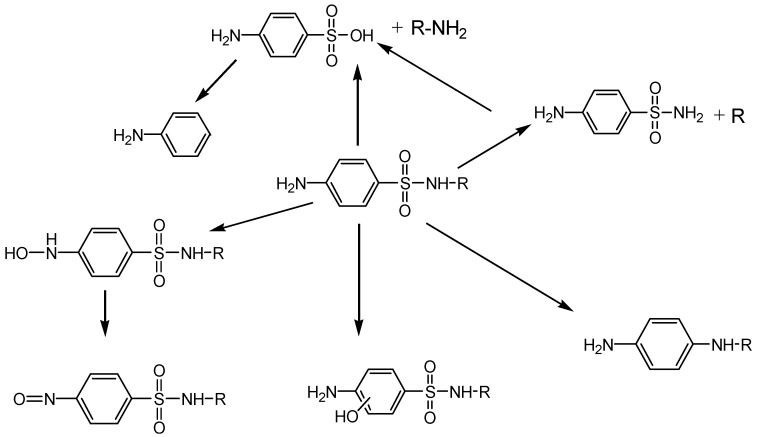
Proposed degradation pathways of sulfonamides during the UV/H_2_O_2_ oxidation process (R represents a heterocyclic ring).

**Table 1 ijerph-16-01797-t001:** Chemical structures and relevant data for selected sulfonamides.

Compound Name	Sulfamethoxazole	Sulfathiazole	Sulfamethazine
Acronym	SMZ	STZ	SMT
Molecular formula	C_10_H_11_N_3_O_3_S	C_9_H_9_N_3_O_2_S_2_	C_12_H_14_N_4_O_2_S
Molecular weight (g/mol)	253.3	255.3	278.3
Pk_a1_	1.6 ± 0.2	2.2 ± 0.1	2.6 ± 0.2
Pk_a2_	5.7 ± 0.2	7.2 ± 0.4	8 ± 1
Molecular structure	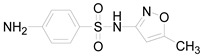	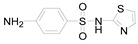	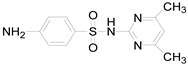
Chemical speciation	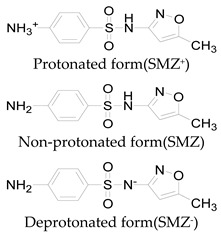	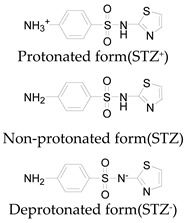	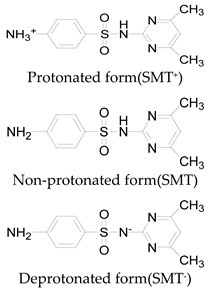
Reference	[16,21]	[7]	[27]

**Table 2 ijerph-16-01797-t002:** The structure of transformation products and their MS fragmentation data in the UV/H_2_O_2_ oxidation process of SMZ.

Name	Retention Time (min)	Molecular Weight (MW)	Characteristic Ions	Molecular Structure
S1	1.666	189	190.0182,172.0045, 122.0243,109.0524	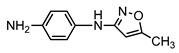
S2	1.713	132	133.0609,72.0449	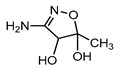
S3	2.615	287	288.0661,156.0119, 108.0452,92.0504	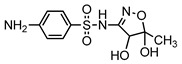
S4	2.843	98	99.0562,72.0462	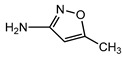
S5	2.785	215	216.0437,156.0113, 108.0444,92.0507, 65.0395	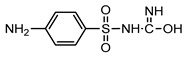
S6	4.049	269	270.0556,172.0065, 124.0398,109.0528	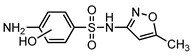
S7	2.882	269	270.0552,156.0116, 108.0446,92.0504	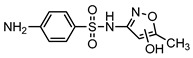
S8	4.184	253	254.0608,156.0120, 108.0451,92.0505	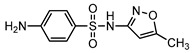

**Table 3 ijerph-16-01797-t003:** The structure of transformation products and their MS fragmentation data in the UV/H_2_O_2_ oxidation process of SMT.

Name	Retention Time (min)	Molecular Weight (MW)	Characteristic Ions	Molecular Structure
S1	2.033	139	140.0824,123.0558, 99.0563,82.0302	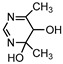
S2	2.160	123	124.0878,107.0610,67.0314	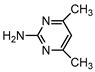
S3	2.965	214	215.1301,108.0689,198.1035	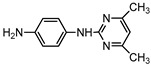
S4	3.812	294	295.0875,124.0874, 186.0336,108.0450,172.0067	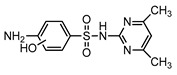
S5	3.218	292	293.0718,212.0823,184.0876	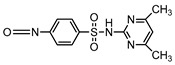
S6	3.984	278	279.0934,186.0347,124.0882, 108.0458,92.0512,156.0126	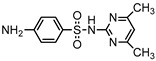
S7	4.313	312	313.0537,186.0336, 142.0052,124.0864	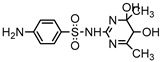

**Table 4 ijerph-16-01797-t004:** The structure of transformation products and their MS fragmentation data in the UV/H_2_O_2_ oxidation process of STZ.

Name	Retention Time (min)	Molecular Weight (MW)	Characteristic Ions	Molecular Structure
S1	1.692	100	101.0179,74.0078,58.9978	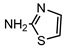
S2	2.208	225	226.0662,124.0222, 151.0334	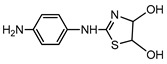
S3	2.483	172	173.0318,93.0582	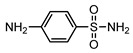
S4	2.648	289	290.0274,156.0120, 108.0452,92.0507	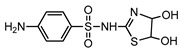
S5	3.423	271	272.0176,172.0069, 124.0400,108.0453	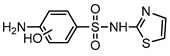
S6	3.864	255	256.0217,156.0120, 108.0452,92.0506	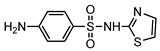
S7	3.900	269	270.0017,189.0123, 161.0174	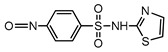

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
