# Peer review of "Removal of Sulfamethoxazole, Sulfathiazole and Sulfamethazine in their Mixed Solution by UV/H2O2 Process"

_ijerph, 2019, doi:10.3390/ijerph16101797_

Round 1
Reviewer 1 Report
Reviewer's comments:
1. Describe the dependencies shown in Figure 1b, 1c and 1d. For example, Figure 1b shows a clear difference in time from 0 to 12 minutes and from 12 minutes to 15 minutes.
2. The same note applies to Figure 2a, 2b and 2c. The relationship should be explained in the text.
3. The same note applies to Figure 3a, 3b and 3c.
4. It should be given what test the linear relationships shown in Figure 4d were tested. The correlation coefficient of the analyzed 3 dependencies should be provided.
The work presents interesting research results, however, the work should be supplemented with the above remarks
Author Response
Point 1: Describe the dependencies shown in Figure 1b, 1c and 1d. For example, Figure 1b shows a clear difference in time from 0 to 12 minutes and from 12 minutes to 15 minutes.
Response 1: Thank you for your great comments and suggestions. In the analytical method we mentioned that the pseudo-first-order kinetic model was used to fit the degradation data of sulfonamides. The ln(C0/Ct) as a function of the time (min) is plotted and the degradation rate constant can be obtained by the linear relationship of the graphs. Here we plotted the degradation rate constant of each compound at different oxidation treatment (Fig. 1e). This Figure 1 was replaced with a new one. The new Figure 1 is shown below. The figure captions of Figure 1 have been modified. The modifications were shown as follows.
We added the sentences “The degradation rate constant of SMZ, STZ and SMT was found to be 0.155 min-1, 0.044 min-1 and 0.016 min-1, respectively, when there was only sole UV irradiation in the system (Figure 1e).” and “At this time, the degradation rate constant of SMZ, STZ and SMT increased to 0.276 min-1, 0.178 min-1 and 0.130 min-1, respectively (Figure 1e).” in the text. All revised parts were marked in red text. We hope our response will make you feel satisfied.
Figure 1. UV spectrum scans of sulfonamides (a), degradation of SMZ (b), STZ (c) and SMT (d) in different oxidation treatments, and degradation rate constants for sulfonamides degradation at different oxidation treatment (e). Reaction conditions: initial concentration of the mixed solution 100 μg L-1 (including SMZ 100 μg L-1, STZ 100 μg L-1 and SMT 100 μg L-1), UV light intensity 5 W, H2O2 dosage 55 mg L-1
Point 2: The same note applies to Figure 2a, 2b and 2c. The relationship should be explained in the text.
Response 2: Thank you for your great comments and suggestions. Similarly, we plotted the degradation rate constant of each compound at different initial concentration of the mixed solution (Fig. 2d). This Figure 2 was replaced with a new one. The new Figure 2 is shown below. The figure captions of Figure 2 have been modified. The modifications were shown as follows. All revised parts were marked in red text. We hope our response will make you feel satisfied.
Figure 2. Effect of initial concentration of the mixed solution on the removal of SMZ (a), STZ (b) and SMT (c) and degradation rate constants for sulfonamides degradation at different initial concentration of the mixed solution (d). Reaction conditions: UV light intensity 5 W, H2O2 dosage 55 mg L-1
Point 3: The same note applies to Figure 3a, 3b and 3c.
Response 3: Thank you for your great comments and suggestions. Similarly, we plotted the degradation rate constant of each compound at different UV light intensity (Fig. 3d). This Figure 3 was replaced with a new one. The new Figure 3 is shown below. The figure captions of Figure 3 have been modified. The modifications were shown as follows. All revised parts were marked in red text. We hope our response will make you feel satisfied.
Figure 3. Effect of UV light intensity on the removal of SMZ (a), STZ (b) and SMT (c) and degradation rate constants for sulfonamides degradation at different UV light intensity (d). Reaction conditions: initial concentration of the mixed solution 100 μg L-1, H2O2 dosage 55 mg L-1
Point 4: It should be given what test the linear relationships shown in Figure 4d were tested. The correlation coefficient of the analyzed 3 dependencies should be provided.
Response 4: Thank you for your great comments and suggestions. The linear relationship is fitting the degradation data of sulfonamides at different H2O2 dosage according to Equation 2 shown in Section 2.3. The degradation rate constant is obtained by the slope of the linear regression of the sulfonamides degradation data points. Then the degradation rate constant of each compound as a function of different H2O2 dosage is plotted and the Figure 4d is obtained. In order to make the fitting data of each compound clearer, we added sentences “The degradation of sulfonamides at different H2O2 dosage fitted the pseudo-first-order kinetic models based on the Equation 2 and this model had a good fitness (R2 > 0.94) with the degradation of sulfonamides. When the H2O2 dosage was 0, 30, 55, and 100 mg L-1, the degradation rate constant of SMZ, STZ and SMT was 0.139, 0.171, 0.197, and 0.266 min-1, 0.039, 0.073, 0.137, and 0.181 min-1, and 0.014, 0.063, 0.101, and 0.158 min-1, respectively.” in the text. All revised parts were marked in red text. We hope our response will make you feel satisfied.

Reviewer 2 Report
The manuscript describes the photocatalyzed degradation (UV/H2O2) in water of sulfamethoxazole (SMZ), sulfathiazole (STZ) and sulfamethazine (SMT) sulfonamide antibiotics in their mixed solution.
Sulfonamide antibiotics may be completely transformed in the presence of the other contaminants following the pseudo-first-order kinetic model.
Removal of sulfonamide antibiotics was dependent on the initial concentration of mixed solution, the intensity of UV light irradiation, the dosage of H2O2 and the initial pH of the solution. Increase of UV light intensity and H2O2 concentration substantially improved the decomposition efficiency, while a higher initial concentration of mixed solution heavily suppressed the decomposition rate. The effect of pH was also discussed.
The photodegradation intermediates of SMZ, STZ and SMT during the UV/H2O2 process were identified and a corresponding degradation pathway was proposed, although for each individual antibiotic and not for the mixed solution. Results are in good agreement with previous research on these compounds.
The manuscript lacks novelty (technical note?) and the mechanistic proposal is for each antibiotic and not for the mixture. Minor changes would be required before publication, if this is the case:
Keywords should be revised since some of them are too general and provide little information, e.g., mixed solution, intermediates. Photodegradation and AOP are suggested.
Text in the 2nd paragraph of the Introduction should be revised for clarity:
Sulfonamide (antibiotics)...
Concentrations should be given in Figure 1 caption. Figure 1A should show the absorption coefficient (epsilon) instead of Absorbance, in particular, to support the statement in line 139 regarding the stronger light absorption (254 nm) by SMZ, as well as the discussion in lines 149-161.
Line 169-170. The concentrations reported are for each antibiotic or for the mixture, please clarify this issue.
Lines 181-182 sulfonamide and not sulfonamides.
Eq (4) OH species, radical or anion?
Lines 245 and below. For the sake of clarity, a table should be given including the pKa values ant the corresponding protonated/neutral/deprotonated species. This would make the discussion quite more comprehensive.
Line 255 Absorption coefficient, and not extinction coefficient (obsolete).
Line 308 Loss, instead of “deprotection”?
Subindexes oh H2O2 in Tables 1-3 headings, and in Figure 6 structure for the alkyl amine.
Author Response
Point 1: The manuscript lacks novelty (technical note?) and the mechanistic proposal is for each antibiotic and not for the mixture. Minor changes would be required before publication, if this is the case:
Keywords should be revised since some of them are too general and provide little information, e.g., mixed solution, intermediates. Photodegradation and AOP are suggested.
Response 1: Thank you for your great comments and suggestions. In this paper, our experimental design is based on the mixed solution. In the experiment, we added the mixed solution of SMZ, STZ and SMT with trace concentration in the reactor. Due to the retention time of these three compounds is different during the HPLC tests, we can simultaneously measure the concentration of these three compounds based on the measured standard curve of each standard compound. This achieves the purpose of simultaneously removing the three compounds in the mixed solution. According to the investigated literature, the study about the simultaneous removal of antibiotics with trace concentration in the mixed solution using UV/H2O2 process is quite limited. This is also the novelty of this article.
In addition, we used “Keywords: Sulfonamides; Mixed solution; UV/H2O2 oxidation; Photodegradation; Advanced oxidation process” to replace “Keywords: Sulfonamide antibiotic; Mixed solution; UV/H2O2 oxidation; Degradation; Intermediates” All revised parts were marked in red text. We hope our response will make you feel satisfied.
Point 2: Text in the 2nd paragraph of the Introduction should be revised for clarity:
Sulfonamide (antibiotics).
Response 2: Thank you for your great comments and suggestions. We revised the 2nd paragraph to “Sulfonamide is a collection of important synthetic sulfanilamide derivatives and is widely used in human and veterinary medicine [3–5]. It is reported that there are eight common sulfonamides including sulfamethoxazole, sulfamethizole, sulfadiazine, sulfacetamide, sulfisoxazole, sulfanilamide, sulfasalazine and sulfadoxine [7]. In this work, sulfathiazole (STZ), sulfamethoxazole (SMZ) and sulfamethazine (SMT) were chosen as the target because that: (a) given that these molecules are polar, amphoteric, water-soluble substances with light and thermal stability [1,6], they possess high migration ability and can easily and quickly spread in the environment [1,6]; (b) SMZ, SMT and STZ are the most heavily reported drugs in the surface water of China and exhibit a high pollution level [8-10]; and (c) all the sulfonamides are found in the wastewater treatment plant effluent, groundwater, surface water and even drinking water supply [5,10]. The detection of sulfonamides in the treated drinking water and wastewater treatment plant effluent indicates that conventional water and wastewater treatment processes cannot effectively eliminate sulfonamides. For these aforementioned reasons, it is highly desired to develop reliable water treatment methods that can efficiently remove sulfonamides at a trace level, especially in China.” All revised parts were marked in red text. We hope our response will make you feel satisfied.
Point 3: Concentrations should be given in Figure 1 caption. Figure 1A should show the absorption coefficient (epsilon) instead of Absorbance, in particular, to support the statement in line 139 regarding the stronger light absorption (254 nm) by SMZ, as well as the discussion in lines 149-161.
Response 3: Thank you for your great comments and suggestions. We added the experimental conditions in Figure 1 captions. The figure captions of Figure 1 have been modified. The modifications were shown as follows:
Figure 1. UV spectrum scans of sulfonamides (a), degradation of SMZ (b), STZ (c) and SMT (d) in different oxidation treatments, and degradation rate constants for sulfonamides degradation at different oxidation treatment (e). Reaction conditions: initial concentration of the mixed solution 100 μg L-1 (including SMZ 100 μg L-1, STZ 100 μg L-1 and SMT 100 μg L-1), UV light intensity 5 W, H2O2 dosage 55 mg L-1
We used “Absorption coefficient (ɛ)” to replace “Absorbance” in Figure 1a and this Figure 1 was replaced with a new one. The new Figure 1 is shown below. In addition, the statement in line 139 about the stronger light absorption of SMZ is not obvious in the Figure 1a. In order to avoid the appearance of this uncertainly, we deleted this sentence in line 139. All revised parts were marked in red text. We hope our response will make you feel satisfied.
Point 4: Line 169-170. The concentrations reported are for each antibiotic or for the mixture, please clarify this issue.
Response 4: Thank you for your great comments and suggestions. The concentration of the solution mentioned in this article is the concentration of the mixed solution. In the experiment, we added the mixed solution of SMZ, STZ and SMT with a certain concentration in the reactor. The retention time of these three compounds is different during the HPLC tests. Thus we can simultaneously measure the concentration of these three compounds based on the measured standard curve of each standard compound. This achieves the purpose of simultaneously removing the three compounds in the mixed solution. For the sake of clarity, we have plotted the degradation of each compound separately. We hope our response will make you feel satisfied.
Point 5: Lines 181-182 sulfonamide and not sulfonamides.
Response 5: Thank you for your great comments and suggestions. We used “sulfonamides” to replace “sulfonamides antibiotics” in the full text. All revised parts were marked in red text. We hope our response will make you feel satisfied.
Point 6: Eq (4) OH species, radical or anion?
Response 6: Thank you for your great question. The OH species in the Eq (4) should be the OH anion. In the Eq (4), the OH has been replaced by the OH-. All revised parts were marked in red text. We hope our response will make you feel satisfied.
Point 7: Lines 245 and below. For the sake of clarity, a table should be given including the pKa values and the corresponding protonated/neutral/deprotonated species. This would make the discussion quite more comprehensive.
Response 7: Thank you for your great comments and suggestions. We added a new table. The new Table 1 is shown below. In addition, we used “The pKa values of SMZ, STZ and SMT were shown in Table 1.” to replace the “The pKa values of SMZ, STZ and SMT were 1.7 and 5.6, 2.0 and 7.1, 2.7 and 7.4, respectively.” All revised parts were marked in red text. We hope our response will make you feel satisfied.
Table 1. Chemical structures and relevant data for selected sulfonamides
Compound name | Sulfamethoxazole | Sulfathiazole | Sulfamethazine |
Acronym | SMZ | STZ | SMT |
Molecular formula | C10H11N3O3S | C9H9N3O2S2 | C12H14N4O2S |
Molecular weight (g/mol) | 253.3 | 255.3 | 278.3 |
Pka1 | 1.6±0.2 | 2.2±0.1 | 2.6±0.2 |
Pka2 | 5.7±0.2 | 7.2±0.4 | 8±1 |
Molecular structure | |||
Chemical speciation | Protonated form(SMZ+) Non-protonated form(SMZ) Deprotonated form(SMZ-) | Protonated form(STZ+) Non-protonated form(STZ) Deprotonated form(STZ-) | Protonated form(SMT+) Non-protonated form(SMT) Deprotonated form(SMT-) |
Reference | 16, 21 | 7 | 27 |
Point 8: Line 255 Absorption coefficient, and not extinction coefficient (obsolete).
Response 8: Thank you for your great comments and suggestions. We used “absorption coefficient” to replace “extinction coefficient” in line 255. All revised parts were marked in red text. We hope our response will make you feel satisfied.
Point 9: Line 308 Loss, instead of “deprotection”?
Response 9: Thank you for your great comments and suggestions. We used “loss” to replace “deprotection” in line 308. All revised parts were marked in red text. We hope our response will make you feel satisfied.
Point 10: Subindexes oh H2O2 in Tables 1-3 headings, and in Figure 6 structure for the alkyl amine.
Response 10: Thank you for your great comments and suggestions. We have corrected “H2O2” into “H2O2” in Table 1-3 headings. We also have corrected “R-NH2” to “R-NH2” in Figure 6. All revised parts were marked in red text. We hope our response will make you feel satisfied.

Reviewer 3 Report
The paper contain new interesting data and new aspects e.g. (degradation products inclusion) and fits very well to IJERPH aims, however it needs some improvements before final accepting for publishing
Abstract:
sulfonamides are often referred to as antibiotics, but it is not very precise, in fact term antibiotic reffers to substances naturally occuring with antimicrobial activity whereas antibacterials/antimicrobials (such as sulfonamides) are fully synthetic compounds
In my opinion the use of "antibiotic" towards sulfonamides is not a big mistake (at least in environmental chemistry) , however I suggest to try to avoid "sulfonamide antibiotic" and use word antimicrobials or simply sulfonamides whenever possible
Line 31 pseudo-persistent (in general antibiotics as chemicals are not persistent however due to continuous discharge to the environment can be regarded as pseudo-persistent).
Line 61 – the sentence should not start from “and”
Materials and methods – some more detailed description is missing in case of chemical analyses,
Was any control used ? (reactor with mixture of the three compounds/no treatment(nor UV nor H2O2),
From line 92 –The volume of sample of water is not reported (important to know if sensitivity of method is sufficient), how the target compounds were extracted from water samples (also reference paper should appear- or authors developed their own method?), characteristics of method is not provided (detection limit, quantification limit, precision, accuracy) n=?
Line 122- authors are presenting equation for removal efficiencies but removal does not appear on graphs and in further text, instead degradation rate is used – is this the same?, if yes, removal efficiency in M&M should be replaced by degradation rate
Line 127 “k” in equation 2 is not explained
Line 145 I prefer to replace sulfa antibiotics by sulfonamides
Line 160 consistent or inconsistent?
Line 173 - the sentence should not start from “and”
Figure 2 and text lines 164-174 – in Figure 2 the order of presenting the results is SMZ (a), STZ (b) SMT 9c) while in the text the authors start with STZ and conclude that “similar results were obtained when it comes to SMZ and SMT”, maybe start figure 2 with STZ (will be more consistent with text).
Refer to fig 2 in the text (“similar results were obtained when it comes to SMZ (fig 2a) and SMT 9fig. 2b).
Line 166 “Obviously, the degradation efficiency of STZ decreased with the increase of the 167 initial concentration of mixed solution in the range of test concentrations” – when it is so obvious so why it was investigated (delete “obviously”)
Line 194 the sentence should not start from “and”
From line 187 concentration of hydrogen peroxide and concentrations of sulfonamide mixture should be reported for experiments on effect of UV Light Intensity on Degradation Efficiency (in the text and in figure 3 description)
The same remark for the following subchapter 3.5 (the intensity of the UV light applied and concentrations of sulfonamide mixture should be reported)
Same remark for subchapter 3.6
Line 214 this sentence is not clear – why percentage is used for rate constants changes, refer to fig. 4d (by the way - , decimal numbers are not necessary for 1043.5 ) " Meanwhile, the degradation rate constants of STZ, SMZ and SMT increased by 357.2%, 91.6% and 1043.5%, respectively"
Line 236 The degradation rate constant of STZ increased by 29.3%, why rate constant? I guess the authors mean degradation rate (not constant)
Line 270 “Namely, sulfonamides possessed the delocalized π electrons and were subjected to the pH” not clear – I guess to pH change?
Line 287 ratio
Line 313 “the oxidation position of STZ” what does it mean?
Table 1,2,3 headings “in the UV/H2O2 process of SMZ” - I guess the authors meant “in the UV/H2O2 oxidation process of SMZ”
Line 323 as shown on figure 6
In case of intermediate degradation products – maybe the authors could comment in few sentences their importance, stability/possible environmental/ecological consequences of their formation etc. (in subchapter 3.7 and in conclusions) (it is important for recommendation of UV/H2O2 oxidation process as a good method for removal of organic compounds like sulfonamides)
Author Response
Point 1: Sulfonamides are often referred to as antibiotics, but it is not very precise, in fact term antibiotic refers to substances naturally occuring with antimicrobial activity whereas antibacterials/antimicrobials (such as sulfonamides) are fully synthetic compounds
In my opinion the use of “antibiotic” towards sulfonamides is not a big mistake (at least in environmental chemistry), however I suggest to try to avoid “sulfonamide antibiotic” and use word antimicrobials or simply sulfonamides whenever possible
Response 1: Thank you for your great comments and suggestions. We used “sulfonamides” to replace “sulfonamide antibiotic” in the full text. All revised parts were marked in red text. We hope our response will make you feel satisfied.
Point 2: Line 31 pseudo-persistent (in general antibiotics as chemicals are not persistent however due to continuous discharge to the environment can be regarded as pseudo-persistent).
Response 2: Thank you for your great comments and suggestions. We used “pseudo-persistent” to replace “persistent” in line 31. All revised parts were marked in red text. We hope our response will make you feel satisfied.
Point 3: Line 61 – the sentence should not start from “and”
Response 3: Thank you for your great comments and suggestions. We removed the “and” in this sentence. We hope our response will make you feel satisfied.
Point 4: Materials and methods – some more detailed description is missing in case of chemical analyses,
Response 4: Thank you for your great comments and suggestions. We added sentences “The formic acid used in the high-performance liquid chromatography was guaranteed reagent.” All revised parts were marked in red text. We hope our response will make you feel satisfied.
Point 5: Was any control used? (reactor with mixture of the three compounds/no treatment (nor UV nor H2O2)
Response 5: Thank you for your great question on our manuscript. The control experiments we used included the reactor with mixture of the three compounds with UV irradiation (no H2O2 was added into the reactor) and reactor with mixture of the three compounds with H2O2 addition (without UV irradiation). The main purpose of this experiment was to consider the degradation of contaminants by UV/H2O2 system. The contaminants selected in this article have light and thermal stability so that the possibility of degradation of the contaminants itself in the reactor is very small without the addition of UV and H2O2. Therefore, we did not use the control experiment that the reactor with mixture of the three compounds with neither UV radiation nor H2O2 addition. We hope our response will make you feel satisfied.
Point 6: From line 92-The volume of sample of water is not reported (important to know if sensitivity of method is sufficient), how the target compounds were extracted from water samples (also reference paper should appear- or authors developed their own method?), characteristics of method is not provided (detection limit, quantification limit, precision, accuracy) n=?
Response 6: Thank you for your great question and suggestions. In the experiments, the water sample containing a mixture of the three compounds was added to the reactor and then the water samples were collected at set time intervals. The collected samples were directly used for the determination of residual target compounds by high performance liquid chromatography. During the concentration determination, the pre-prepared target compounds of different concentrations were used as the standard curve and the concentration of residual target compounds was calculated according to the standard curve.
We added the sentences “The initial concentration of the mixed solution of 100 μg L-1 was set.” and “All experiment was performed three times with independent replication of data (n=3).’ in the subchapter 2.2.
We also added the sentence “Under this test condition, the detection limit of the selected sulfonamides was 5 μg L-1, and the relative standard deviations (RSD) of the three substances were all less than 0.2% to meet the detection requirements.” in the subchapter 2.3. All revised parts were marked in red text. We hope our response will make you feel satisfied.
Point 7: Line 122-authors are presenting equation for removal efficiencies but removal does not appear on graphs and in further text, instead degradation rate is used-is this the same?, if yes, removal efficiency in M&M should be replaced by degradation rate
Response 7: Thank you for your great question and suggestions. The removal efficiency and degradation rate mentioned in the text are the same. We used “degradation rate” to replace “removal efficiency” in Materials and Methods. All revised parts were marked in red text. We hope our response will make you feel satisfied.
Point 8: Line 127 “k” in equation 2 is not explained
Response 8: Thank you for your great comments and suggestions. We added the sentences in this section “k is the pseudo-first-order degradation rate constant which is obtained by the slope of the linear regression of the sulfonamides degradation data points.” All revised parts were marked in red text. We hope our response will make you feel satisfied.
Point 9: Line 145 I prefer to replace sulfa antibiotics by sulfonamides
Response 9: Thank you for your great comments and suggestions. We used “sulfonamides” to replace “sulfa antibiotics” in line 145. All revised parts were marked in red text. We hope our response will make you feel satisfied.
Point 10: Line 160 consistent or inconsistent?
Response 10: Thank you for your great questions. We found that the stronger light absorption of SMZ in Figure 1a is not obvious compared with SMT and STZ. Therefore, in order to avoid the appearance of this uncertainly, we delete these two sentences in line 139 and 160. We hope our response will make you feel satisfied.
Point 11: Line 173 - the sentence should not start from “and”
Response 11: Thank you for your great comments and suggestions. We turned the sentence in line 173 into “Similar results were obtained when it comes to STZ degradation (Figure 2b) and SMT degradation (Figure 2c).” All revised parts were marked in red text. We hope our response will make you feel satisfied.
Point 12: Figure 2 and text lines 164-174 – in Figure 2 the order of presenting the results is SMZ (a), STZ (b) SMT (c) while in the text the authors start with STZ and conclude that “similar results were obtained when it comes to SMZ and SMT”, maybe start figure 2 with STZ (will be more consistent with text).
Response 12: Thank you for your great comments and suggestions. We adjust the text statement to “The degradation efficiency of SMZ decreased with the increase of the initial concentration of mixed solution in the range of test concentrations (Figure 2a). The SMZ degradation rate was 99.9% for 50 μg L-1 mixed solution after 20 min of reaction, which decreased to 97.4% and 76.6% when the mixed solution concentration was increased to 100 μg L-1 and 200 μg L-1, respectively. The degradation pattern fitted the pseudo-first-order kinetic model, and the kinetic constant decreased from 0.352 min-1 to 0.197 min-1 and 0.076 min-1 respectively, when the initial concentration of the mixed solution increased from 50 μg L-1 to 100 μg L-1 and 200 μg L-1. Similar results were obtained when it comes to STZ degradation (Figure 2b) and SMT degradation (Figure 2c).” so that the order of presenting the results in Figure 2 is consistent with text. All revised parts were marked in red text. We hope our response will make you feel satisfied.
Point 13: Refer to fig 2 in the text (“similar results were obtained when it comes to SMZ (fig 2a) and SMT (fig. 2b).
Response 13: Thank you for your great comments and suggestions. We used the sentence “Similar results were obtained when it comes to STZ degradation (Figure 2b) and SMT degradation (Figure 2c).” to replace “similar results were obtained when it comes to SMZ and SMT” in the text. All revised parts were marked in red text. We hope our response will make you feel satisfied.
Point 14: Line 166 “Obviously, the degradation efficiency of STZ decreased with the increase of the initial concentration of mixed solution in the range of test concentrations” – when it is so obvious so why it was investigated (delete “obviously”)
Response 14: Thank you for your great comments and suggestions. We removed the “obviously” in this sentence. We hope our response will make you feel satisfied.
Point 15: Line 194 the sentence should not start from “and”
Response 15: Thank you for your great comments and suggestions. We removed the “and” in this sentence in line 194. We hope our response will make you feel satisfied.
Point 16: From line 187 concentration of hydrogen peroxide and concentrations of sulfonamide mixture should be reported for experiments on effect of UV Light Intensity on Degradation Efficiency (in the text and in figure 3 description)
Response 16: Thank you for your great comments and suggestions. We added sentences “The experiments were carried out at UV light intensity of 5 W and the H2O2 dosage of 55 mg L-1.” and “The experiments were carried out at initial concentration of the mixed solution of 100 μg L-1 and the H2O2 dosage of 55 mg L-1.” in the subchapter 3.3 and 3.4, respectively.
We also added the experimental conditions in figure captions. The figure captions of Figure 2-3 have been modified. All the modifications were shown as follows:
Figure 2. Effect of initial concentration of the mixed solution on the removal of SMZ (a), STZ (b) and SMT (c) and degradation rate constants for sulfonamides degradation at different initial concentration of the mixed solution (d). Reaction conditions: UV light intensity 5 W, H2O2 dosage 55 mg L-1
Figure 3. Effect of UV light intensity on the removal of SMZ (a), STZ (b) and SMT (c) and degradation rate constants for sulfonamides degradation at different UV light intensity (d). Reaction conditions: initial concentration of the mixed solution 100 μg L-1, H2O2 dosage 55 mg L-1
In addition, we also added the sentences “The pH of other water samples was not adjusted except for the experiments focusing on the effect of pH on the sulfonamides degradation.” in the subchapter 2.2. All revised parts were marked in red text. We hope our response will make you feel satisfied.
Point 17: The same remark for the following subchapter 3.5 (the intensity of the UV light applied and concentrations of sulfonamide mixture should be reported)
Response 17: Thank you for your great comments and suggestions. We added sentences “Besides, the initial concentration of the mixed solution and the UV light intensity was set as 100 μg L-1 and 5 W, respectively.” in the subchapter 3.5.
We also added the experimental conditions in figure captions. The figure captions of Figure 4 have been modified. The modifications were shown as follows: “Figure 4. Effect of H2O2 dosage on the removal of SMZ (a), STZ (b) and SMT (c) and reaction rate constant as a function of H2O2 dosage (d). Reaction conditions: initial concentration of the mixed solution 100 μg L-1, UV light intensity 5 W” All revised parts were marked in red text. We hope our response will make you feel satisfied.
Point 18: Same remark for subchapter 3.6
Response 18: Thank you for your great comments and suggestions. We added sentences “Besides, the initial concentration of the mixed solution, UV light intensity and H2O2 dosage was set as 100 μg L-1, 5 W and 55 mg L-1, respectively. ” in the subchapter 3.6.
We also added the experimental conditions in figure captions. The figure captions of Figure 5 have been modified. The modifications were shown as follows: “Figure 5. Degradation of SMZ (a), STZ (b) and SMT (c) in different pH and UV spectra of SMZ (d), STZ (e) and SMT (f) under different pH conditions. Reaction conditions: initial concentration of the mixed solution 100 μg L-1, UV light intensity 5 W, H2O2 dosage 55 mg L-1” All revised parts were marked in red text. We hope our response will make you feel satisfied.
Point 19: Line 214 this sentence is not clear – why percentage is used for rate constants changes, refer to fig. 4d (by the way - , decimal numbers are not necessary for 1043.5 ) " Meanwhile, the degradation rate constants of STZ, SMZ and SMT increased by 357.2%, 91.6% and 1043.5%, respectively"
Response 19: Thank you for your great question. In order to more intuitively illustrate the trend that the degradation efficiency of sulfonamides also increases significantly with the substantial increase of H2O2 dosage, we calculate the increase degree of the removal rate constant. With SMZ as example, the removal rate constant of SMZ increased by 91.6% from 0.139 min-1 to 0.266 min-1 when the H2O2 dosage was increased from 0 mg L-1 to 100 mg L-1. For the sake of clarity, we added sentences “The degradation of sulfonamides at different H2O2 dosage fitted the pseudo-first-order kinetic models based on the Equation 2 and this model had a good fitness (R2 > 0.94) with the degradation of sulfonamides. When the H2O2 dosage was 0, 30, 55, and 100 mg L-1, the degradation rate constant of SMZ, STZ and SMT was 0.139, 0.171, 0.197, and 0.266 min-1, 0.039, 0.073, 0.137, and 0.181 min-1, and 0.014, 0.063, 0.101, and 0.158 min-1, respectively. It could be seen that the degradation rate constants of STZ, SMZ and SMT increased by 357.2%, 91.6% and 1043.5%, respectively, when the H2O2 concentration increased from 0 mg L-1 to 100 mg L-1.” in the text. All revised parts were marked in red text. We hope our response will make you feel satisfied.
Point 20: Line 236 The degradation rate constant of STZ increased by 29.3%, why rate constant? I guess the authors mean degradation rate (not constant)
Response 20: Thank you for your great question. Here, this 29.3% mainly refers to the change in the removal rate constant. In order to make the meaning of this sentence clearer, we adjusted this sentence to “The degradation rate constant of STZ increased by 29.3% from 0.120 min-1 to 0.155 min-1 when the initial solution pH increased from 5.0 to 9.0.” All revised parts were marked in red text. We hope our response will make you feel satisfied.
Point 21: Line 270 “Namely, sulfonamides possessed the delocalized π electrons and were subjected to the pH” not clear – I guess to pH change?
Response 21: Thank you for your great question and suggestion. We used sentences “Namely, sulfonamides possessed the delocalized π electrons and were subjected to the pH change” to replace “Namely, sulfonamides possessed the delocalized π electrons and were subjected to the pH” in line 270. All revised parts were marked in red text. We hope our response will make you feel satisfied.
Point 22: Line 287 ratio
Response 22: Thank you for your great comments and suggestions. We used “ratio” to replace “ration” in the text. The revised parts were marked in red text. We hope our response will make you feel satisfied.
Point 23: Line 313 “the oxidation position of STZ” what does it mean?
Response 23: Thank you for your great question. This sentence means that the transformation products of STZ mainly occured in the benzene ring, S-N bond, amino group and thiazole ring. In order to make this sentence clearer, we used sentence “Similarly, the degradation products of STZ mainly including the S-N bond cleavage product (S1, m/z 101.0179), sulfonyl removed product (S2, m/z 226.0662), carbon-carbon double bond addition product (S4, m/z 290.0274), hydroxylation product (S5, m/z 272.0176) and the amino group oxidation product on benzene ring (S7, m/z 270.0017).” to replace the sentence “Similarly, the oxidation position of STZ mainly occurred in the benzene ring, S-N bond, amino group and thiazole ring.” All revised parts were marked in red text. We hope our response will make you feel satisfied.
Point 24: Table 1,2,3 headings “in the UV/H2O2 process of SMZ” - I guess the authors meant “in the UV/H2O2 oxidation process of SMZ”
Response 24: Thank you for your great comments and suggestions. We used “in the UV/H2O2 oxidation process of SMZ”, “in the UV/H2O2 oxidation process of SMT”, and “in the UV/H2O2 oxidation process of STZ” to replace “in the UV/H2O2 process of SMZ”, “in the UV/H2O2 process of SMT”, and “in the UV/H2O2 process of STZ”, respectively. All revised parts were marked in red text. We hope our response will make you feel satisfied.
Point 25: Line 323 as shown on figure 6
Response 25: Thank you for your great comments and suggestions. We used “as shown on Figure 6” to replace “as Figure 6”. The revised parts were marked in red text. We hope our response will make you feel satisfied.
Point 26: In case of intermediate degradation products – maybe the authors could comment in few sentences their importance, stability/possible environmental/ecological consequences of their formation etc. (in subchapter 3.7 and in conclusions) (it is important for recommendation of UV/H2O2 oxidation process as a good method for removal of organic compounds like sulfonamides)
Response 26: Thank you for your great comments and suggestions. We added sentences “The contaminants present in the aqueous solutions might produce more toxic conversion products than the parent compound. Some researchers evaluated the potential environmental impact of sulfonamides and its byproducts [29-31]. They found that toxicity was higher at the beginning of the reaction and the toxicity reduced gradually with the further progress of reaction. This showed that the intermediates of sulfonamides were degraded into the less toxic products as the reaction progresses. So, the environmental impact of sulfonamides degradation intermediates was less than that before degradation.” in subchapter 3.7.
We added sentences “The toxicity of sulfonamides was high at the beginning, and the toxicity was gradually reduced as the reaction progresses.” in conclusion
We also added the new reference “29. Justo, A.; González, O.; Aceña, J.; Pérez, S.; Barceló, D.; Sans, C.; Esplugas, S. Pharmaceuticals and organic pollution mitigation in reclamation osmosis brines by UV/H2O2 and ozone. J. Hazard. Mater. 2013, 263, 268-274.”, “30. Hou, L.; Zhang, H.; Wang, L.; Chen, L.; Xiong, Y.; Xue, X. Removal of sulfamethoxazole from aqueous solution by sono-ozonation in the presence of a magnetic catalyst. Sep. Purif. Technol. 2013, 117, 46-52.” and “31. Guo, W.Q.; Yin, R.L.; Zhou, X.J.; Cao, H.O.; Chang, J.S.; Ren, N.Q. Ultrasonic-assisted ozone oxidation process for sulfamethoxazole removal: Impact factors and degradation process. Desalin. Water Treat. 2016, 57, 21015-21022.” All revised parts were marked in red text. We hope our response will make you feel satisfied.

Round 2
Reviewer 3 Report
In my opinion the authors significantly improved the quality of the manuscript
I would just suggest to add in analytical methods line 107-111.
the remaining part of response 6 at present not included in the text "The collected water samples were directly used for the determination of residual target compounds by high performance liquid chromatography.
The pre-prepared target compound solutions of different concentrations were used to create the calibration curve, which was employed to calculated the concentration of residual target compounds.
Please consider the correctness of the title : Removal of sulfamethoxazole, sulfathiazole and sulfamethazine from their coexistence solution by UV/H2O2 system
this phrase "from their coexistence solution" sounds a bit strange , but I'm not a native speaker of course.
Author Response
Point 1: I would just suggest to add in analytical methods line 107-111.
the remaining part of response 6 at present not included in the text "The collected water samples were directly used for the determination of residual target compounds by high performance liquid chromatography.
The pre-prepared target compound solutions of different concentrations were used to create the calibration curve, which was employed to calculated the concentration of residual target compounds.
Response 1: Thank you for your great comments and suggestions. We added the sentences “The collected water samples during the experiment were directly used for the determination of residual target compounds by HPLC according to the above conditions. The pre-prepared target compound solution of different concentrations was used to create the calibration curve, which was employed to calculate the concentration of residual target compounds.” in the analytical methods. All revised parts were marked in red text. We hope our response will make you feel satisfied.
Point 2: Please consider the correctness of the title: Removal of sulfamethoxazole, sulfathiazole and sulfamethazine from their coexistence solution by UV/H2O2 system
this phrase "from their coexistence solution" sounds a bit strange , but I'm not a native speaker of course.
Response 2: Thank you for your great comments and suggestions. We have corrected the title “Removal of sulfamethoxazole, sulfathiazole and sulfamethazine from their coexistence solution by UV/H2O2 system” into “Removal of Sulfamethoxazole, Sulfathiazole and Sulfamethazine in their Mixed Solution by UV/H2O2 Process”. All revised parts were marked in red text. We hope our response will make you feel satisfied.
